# Oocyte Maturation in Starfish

**DOI:** 10.3390/cells9020476

**Published:** 2020-02-19

**Authors:** Kazuyoshi Chiba

**Affiliations:** Department of Biological Sciences, Ochanomizu University, Tokyo 112-8610, Japan; chiba.kazuyoshi@ocha.ac.jp; Tel.:+81-3-5978-5370

**Keywords:** starfish, oocyte maturation, SGK, intracellular pH, fertilization

## Abstract

Oocyte maturation is a process that occurs in the ovaries, where an immature oocyte resumes meiosis to attain competence for normal fertilization after ovulation/spawning. In starfish, the hormone 1-methyladenine binds to an unidentified receptor on the plasma membrane of oocytes, inducing a conformational change in the heterotrimeric GTP-binding protein α-subunit (Gα), so that the α-subunit binds GTP in exchange of GDP on the plasma membrane. The GTP-binding protein βγ-subunit (Gβγ) is released from Gα, and the released Gβγ activates phosphatidylinositol-3 kinase (PI3K), followed by the target of rapamycin kinase complex2 (TORC2) and 3-phosphoinositide-dependent protein kinase 1 (PDK1)-dependent phosphorylation of serum- and glucocorticoid-regulated kinase (SGK) of ovarian oocytes. Thereafter, SGK activates Na^+^/H^+^ exchanger (NHE) to increase the intracellular pH (pHi) from ~6.7 to ~6.9. Moreover, SGK phosphorylates Cdc25 and Myt1, thereby inducing the de-phosphorylation and activation of cyclin B–Cdk1, causing germinal vesicle breakdown (GVBD). Both pHi increase and GVBD are required for spindle assembly at metaphase I, followed by MI arrest at pHi 6.9 until spawning. Due to MI arrest or SGK-dependent pHi control, spawned oocytes can be fertilized normally

## 1. Introduction

Since the discovery of 1-methyladenine (1-MA)-induced meiosis resumption of starfish oocytes arresting at prophase of meiosis I (Pro I) [1,2], most experiments have been conducted using isolated oocytes from the animals. However, the isolated oocytes do not always behave the same as the ovarian oocytes in vivo. For instance, the isolated oocytes in seawater do not arrest at the metaphase of meiosis I (MI) after induction of GVBD, whereas the ovarian oocytes in the female animals in vivo undergo MI arrest until spawning or ovulation [3,4,5]. Owing to this, naturally spawned oocytes are in optimum state for monospermy, when only a single sperm fertilizes the egg [6].

Oocyte maturation is a process by which an immature oocyte resumes meiosis to become a fertilizable egg or to attain competence for normal fertilization after ovulation/spawning [7,8,9,10]. Therefore, the study of in vivo maturation of ovarian oocytes as well as in vitro maturation of isolated oocytes is still required. In this review, the mechanisms for induction of starfish oocyte maturation in vivo as well as in vitro are discussed.

## 2. Activation of G-Protein Coupled Receptor by Maturation Inducing Hormone 1-MA

Starfish oocytes are surrounded by follicle cells inside the ovaries. The insulin-like growth factor/relaxin released from the radial nerve stimulates the follicle cells to release the hormone 1-MA [11]. Although the receptor on oocytes for 1-MA has not been identified, binding of the radiolabeled 1-MA to the oocyte membrane has two apparent Kds of approximately 30 nM and more than 1 µM. The high-affinity form is converted into the low-affinity one in the presence of a GTP analogue [12]. These results suggest that the 1-MA receptor on the oocyte plasma membrane binds to the GTP-binding protein. Pertussis toxin injected into the isolated oocytes induces ADP-ribosylation of GTP-binding protein α subunit (Gα), thus leading to GVBD blockage [13,14]. Furthermore, GVBD is induced by injection of starfish G-protein βγ subunits (Gβγ) purified from starfish oocyte [15,16] as well as by mammalian Gβγ [17]. Subsequently, the oocytes become fertilizable after GVBD [16,17]. In addition, Gβγ expression by mRNAs, coding for Gβ and Gγ, can induce GVBD when injected into immature oocytes [18]. Therefore, after the stimulation of the 1-MA receptor on the plasma membrane, Gβγ released from the Gα interacts with effector(s) to induce oocyte maturation in starfish (Figure 1).

The pHi of isolated oocytes is ~7.0, whereas the pHi of the oocytes in the female animals in vivo is ~6.7 [6]. These differences do not cause any delay of signal transduction from 1-MA to G-protein.

## 3. The Effectors of G-Protein

### 3.1. SGK-Dependent GVBD

The v-Akt murine thymoma viral oncogene/protein kinase-B (Akt) pleckstrin homology (PH) domain interacts with phosphatidyl inositol (3, 4, 5) triphosphate (PIP3) [19]. When the PH domain fused with GFP (PH-GFP) is expressed in the isolated oocytes, PH-GFP localizes on the plasma membrane upon 1-MA stimulation [18]. These results suggest that Gβγ stimulates PI3K to produce PIP3. In addition, an inhibitor of PI3K, such as wortmannin, blocks the Gβγ- and 1-MA-dependent GVBD [20,21]. Thus, Gβγ activates PI3K to induce GVBD. However, an unidentified effector of Gβγ may be additionally involved in 1-MA signal transduction, because the sole expression of constitutively active PI3K cannot induce GVBD [18]. Interestingly, GVBD is induced by the simultaneous expression of the constitutively active PI3K and the mutant Gβγ, which is unable to induce GVBD by itself [18]. These results suggest that the wild-type Gβγ has two domains that activate PI3K and the other unknown effector.

Although this unknown effector of Gβγ remains to be determined, these effectors cooperatively activate serum- and glucocorticoid-regulated kinase (SGK) through two kinases, PDK1 and TORC2 (Figure 1); starfish SGK possesses Thr312 in its activation loop and Thr479 in the hydrophobic motif (HM), which are phosphorylated by PDK1 and TORC2, respectively [22,23]. Moreover, phosphorylation of these two amino acids is required for the activation of SGK, because the inhibitors of these two protein kinases block the phosphorylation of SGK, kinase activity of SGK, and 1-MA-dependent GVBD [22,23]. Further, the antibody raised against SGK-HM completely blocks GVBD and inhibits the phosphorylation of Thr312 as well as Thr479 [22]. The specificity of this antibody is confirmed by conducting a rescue experiment using the mutant SGK (T479E), mimicking the phosphorylation of the HM by TORC2. Even in the presence of the antibody, 1-MA induces both the phosphorylation of Thr312 in the mutant protein and GVBD, whereas phosphorylation of the endogenous SGK is blocked [22]. These results indicate that the mutant SGK, which is phosphorylated by PDK1, induces GVBD. Moreover, the activated SGK phosphorylates Cdc25 and Myt1 [23] to activate cyclin B–Cdk1 [2] (Figure 1). Although Akt was previously believed to be involved in the phosphorylation of Cdc25 [18] and Myt1 [24], Akt in vivo cannot activate cyclin B–Cdk1 [23].

### 3.2. SGK-Dependent Spindle Formation

In the coelomic fluid of living animals, the concentration of dissolved CO_2_ is ~1.0%, which is 20 times higher than that in normal air [25]. As CO_2_ can easily enter the plasma membrane, high CO_2_ levels would decrease the pHi. The pHi of ovarian oocytes in vivo has been estimated to be ~6.7, by studying oocytes that are incubated in seawater, where the gas conditions are similar to those of the coelomic fluid of living animals [25]. Under in vivo gas conditions, the 1-MA increases pHi, but the relatively high dissolved CO_2_ in the ovary limits the rise to ~6.9. This is still sufficient to block meiotic progression. Soon after GVBD, contraction of the ovary wall makes the oocytes spawn into sea water. In the lower dissolved CO_2_ of sea water, the pHi rises to ~7.3, which releases the meiotic block [25]. 

In mammalian cells, NHE increases the pHi and regulates the timing of G2/M entry and transition [26]. In starfish, as the NHE-dependent increase in pHi occurs immediately after 1-MA stimulation or Gβγ injection [3,25], the effector of Gβγ should activate NHE. Indeed, starfish NHE has the consensus sequence of SGK-dependent phosphorylation [3], and NHE activation is blocked by the antibody raised against SGK HM [22]. Although an increase in pHi followed by cyclin B–Cdk1 activation is inhibited by the anti-SGK antibody, the activation of cyclin B-Cdk1 is not dependent on the SGK-induced pHi increase. This was shown by clamping the pHi at 6.7 using artificial sea water containing ammonium acetate, a condition which did not block Cdc2/Cdk1 activation [22,25]. In addition, a filamentous actin (F-actin) shell, which is essential for nuclear envelope fragmentation [27,28], is formed on the inner surface of the GV at pHi 6.7 [22].

Moreover, Cdc25 is activated earlier at a lower pHi as compared to that at a higher pHi of isolated oocytes under normal air conditions [22]. These results indicate that the 1-MA signal transduction from 1-MA to cyclin B–Cdk1 activation occurs optimally in vivo at a lower pHi. Thus, the pHi increase is not required for cyclin B–Cdk1 activation.

More importantly, chromosome transport and spindle formation occur in the ovarian oocytes due to an increase in the pHi, because pHi of non-stimulated oocytes in the ovaries are estimated to be ~6.7, at which chromosome transport and spindle formation are blocked [22]. Because chromosomes are transported via contractile flow of the actin meshwork toward the animal pole [27,29], an actin disassembly-driven contractile flow of the meshwork [30] may be sensitive to pHi changes.

Thus, SGK increases pHi as well as the activity of cyclin B–Cdk1-causing GVBD, which cooperatively induces chromosome transport and spindle formation at MI (Figure 1). When the SGK-dependent pHi increase reaches ~7.0, dephosphorylation of SGK occurs [22]. This inactivation of SGK may stop the pHi increase, causing MI arrest at pHi6.9.

## 4. MI Arrest in the Ovaries

The MI arrest is important for the normal fertilization [6]. If this arrest did not occur in the ovaries, many oocytes would be spawned after polar body formation when polyspermy occurs, because the spawning period is too long to release all oocytes before the polar body formation [3,5]. Thus, MI arrest maintains the cell-cycle phase of oocytes, thereby promoting normal fertilization [6]. In addition, ovarian oocytes are physically separated from spawned sperm before GVBD due to external fertilization in starfish, indicating that the ovaries play a role in protecting immature oocytes from polyspermy until they develop the competence for a normal fertilization process [6]. Moreover, insemination of oocytes before stimulation of 1-MA causes polyspermy; this is why oocytes at this stage are called “immature”.

## 5. Oocyte Maturation in Mammals

Here, oocyte maturation in mammals has been briefly summarized to compare it with that of starfish.

Pertussis toxin does not prevent the resumption of meiotic arrest at Pro I in the mouse oocytes [31], but the orphan receptor GPR3 constitutively activates Gs, which increases cAMP and activates cAMP-dependent protein kinase A (PKA) [32], thus maintaining the Cdk1 in an inactive form [33].

When mammalian oocytes are isolated from preovulatory follicles in the ovaries, meiosis resumes spontaneously, indicating that the somatic cells, such as cumulus cells and granulosa cells, surrounding the oocytes in the follicles play a role to block meiosis resumption [34]. Indeed, cyclic GMP (cGMP) produced in the granulosa cells by the guanylyl cyclase natriuretic peptide receptor 2 (NPR2) [35] diffuses through gap junctions into the cumulus cells and the oocytes, which competitively inhibit cAMP phosphodiesterase, PDE3A, thereby maintaining Pro I arrest [36,37]. Instead, the luteinizing hormone (LH) induces cGMP decrease in the granulosa cells and in the oocytes [38,39], which activates PDE3A to decrease cAMP concentration in the oocytes, inducing the activation of Cdk1 kinase and GVBD [36,37].

After the first polar body formation, oocytes arrest at the metaphase stage of meiosis II (MII). Then, they are ovulated from the follicles and enter the oviducts, where fertilization occurs. MII arrest is released by fertilization, followed by the second polar body formation. If oocytes are isolated from the ovary and inseminated prior to the stage of ovulation, polyspermy occurs and the developing embryo dies [40], indicating that ovaries protect immature oocytes from polyspermy [6].

The molecules mainly involved in the release from the Pro I arrest in mammalian oocytes are different from those in starfish, although insulin signaling acts cooperatively with gonadotropins in mammals [41]. More importantly, the presence of Pro I arrest during oogenesis in the animal kingdom is evolutionarily conserved, probably due to an advantage of tetraploid during Pro I arrest rather than haploid after meiotic division to produce a big egg containing maternal factors such as maternal mRNAs.

## 6. Future Directions

Starfish oocyte maturation in vivo is controlled by the two SGK-dependent events i.e., GVBD and pHi increase (Figure 1). These two events regulate the chromosome transport, spindle formation and MI arrest, which are required for normal fertilization.

Because SGK possesses the Phox homology (PX) domain [22], which may interact with PI(3)P, one possible candidate acting as the effector of Gβγ may be SH2-containing inositol phosphatase (SHIP) or inositol polyphosphate-4-phosphatase (INPP4A/B) [42,43,44]. The PIP3 produced by PI3K may get dephosphorylated by SHIP followed by INPP4 A/B, forming PI(3)P on the plasma membrane. This may recruit starfish SGK to the plasma membrane, where SGK is activated by the TORC2 and PDK1. Mutant Gβγ may not activate SHIP or INPP4 A/B. Because PI(3)P is also directly produced by the phosphorylation of PI in early endosomes during intracellular vesicle trafficking in mammalian cells [45], causing activation of NHE [46], starfish SGK may get activated in the early endosomes. Future studies are required to understand the mechanism underlying in vivo oocyte maturation in detail.

## Figures and Tables

**Figure 1 cells-09-00476-f001:**
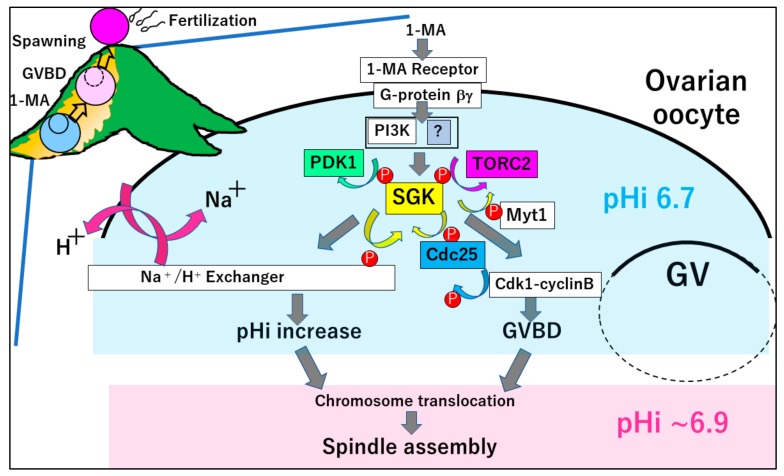
Meiosis resumption of starfish oocytes in the ovaries. The hormone 1-MA binds to an unidentified receptor, to release Gβγ from Gα. The Gβγ activates PI3K, followed by TORC2 and PDK1-dependent phosphorylation of SGK. Then, SGK activates NHE to increase the intracellular pH from ~6.7 to ~6.9. In addition, SGK phosphorylates Cdc25 and Myt1, thereby inducing the activation of cyclin B–Cdk1 and GVBD. Both pHi increase and GVBD are required for the spindle assembly at metaphase I.

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
