# Peer review of "Oocyte Maturation in Starfish"

_cells, 2020, doi:10.3390/cells9020476_

Round 1

Reviewer 1 Report

This is a good review of the signal transduction pathway in meiotic maturation of starfish oocytes. The author has been involved in many of the important papers on this subject in recent years, so it is very appropriate. Two of the papers were published last year in Journal of Cell Biology, so it is timely as well.

I found a couple of places that could be improved for clarity and several smaller english corrections.

The very last part, "Conclusions", is not really conclusions, and maybe could be labeled "Future directions" instead.

Because the isolated oocytes are such a good experimental system, no one had thought to investigate meiosis as it occurs in the animal. It is to the author's credit that he did this, and in the process found evidence for a major role of pH. I felt that the involvement of pH could be more clearly stated. Here is my attempt.
Line 89 and following: 1-MA increases pHi, but the relatively high dissolved CO2 in the ovary limits the rise to ~6.9. This is still sufficient to block meiotic progression. Soon after GVBD, the oocytes induce contraction of the ovary wall, and the oocytes are spawned into sea water. In the lower dissolved CO2 of sea water, the pHi rises to ~7.3, which releases the meiotic block.

line 26 - remove 'the' at the end of the line
line 27 - remove 'the' from 'the oocytes'
line 27 - replace 'in a way similar to that of' with 'the same as'
line 31 - replace 'all oocytes...' with 'naturally spawned oocytes are in optimum state for monospermy'

line 44 - remove 'The' from 'The pertussis'
line 70 - define SGK as 'serum- and glucocorticoid- regulated kinase'

line 98 - replace 'This is due to the fact that...' by 'This was shown by clamping the pHi at 6.7 using artificial sea water containing ammonium acetate, a condition which did not block Cdc2/Cdk1 activation.'

line 113 remove 'to'
line 123 ; this is why oocytes at this stage are called 'immature'.

Author Response

I appreciate suggestions from the reviewer 1. They are very helpful and improve the manuscript.

I replicated the comments of the Reviewer 1, and I wrote answers for the comments.

Comment 1: The very last part, "Conclusions", is not really conclusions, and maybe could be labeled "Future directions" instead.

Answer to comment 1: As suggested, I used "Future directions"

Comment 2: Because the isolated oocytes are such a good experimental system, no one had thought to investigate meiosis as it occurs in the animal. It is to the author's credit that he did this, and in the process found evidence for a major role of pH. I felt that the involvement of pH could be more clearly stated. Here is my attempt.
Line 89 and following: 1-MA increases pHi, but the relatively high dissolved CO2 in the ovary limits the rise to ~6.9. This is still sufficient to block meiotic progression. Soon after GVBD, the oocytes induce contraction of the ovary wall, and the oocytes are spawned into sea water. In the lower dissolved CO2 of sea water, the pHi rises to ~7.3, which releases the meiotic block.

Answer to comment 2: As indicated, I changed line 89-., although I would like to modify only one sentence “the oocytes induce contraction of the ovary wall, and the oocytes are spawned into sea water.”, because I am not sure the oocytes induce contraction of the ovary wall. Then, I would modify it: “contraction of the ovary wall makes the oocytes spawned into sea water”

If this modification is good, final version should be:

“the 1-MA increases pHi, but the relatively high dissolved CO2 in the ovary limits the rise to ~6.9. This is still sufficient to block meiotic progression. Soon after GVBD, contraction of the ovary wall makes the oocytes spawned into sea water. In the lower dissolved CO2 of sea water, the pHi rises to ~7.3, which releases the meiotic block”

Comment 3: line 26 - remove 'the' at the end of the line.

Answer to Comment 3: As indicated, I removed 'the' at the end of the line.

Comment 4: line 27 - remove 'the' from 'the oocytes'

Answer to Comment 4: As indicated, I removed 'the' from 'the oocytes'.

Comment 5: line 27 - replace 'in a way similar to that of' with 'the same as'

Answer to Comment 5: As indicated, I replaced 'in a way similar to that of' with 'the same as'.

Comment 6: line 31 - replace 'all oocytes...' with 'naturally spawned oocytes are in optimum state for monospermy'

Answer to Comment 6: As indicated, I replaced 'all oocytes...' with 'naturally spawned oocytes are in optimum state for monospermy'.

Comment 7: line 44 - remove 'The' from 'The pertussis'

Answer to Comment 7: As indicated, I removed 'The' from 'The pertussis'.

Comment 8: line 70 - define SGK as 'serum- and glucocorticoid- regulated kinase'

Answer to Comment 8: As indicated, I defined SGK as 'serum- and glucocorticoid- regulated kinase'.

Comment 9: line 98 - replace 'This is due to the fact that...' by 'This was shown by clamping the pHi at 6.7 using artificial sea water containing ammonium acetate, a condition which did not block Cdc2/Cdk1 activation.'

Answer to Comment 9: As indicated, I replaced 'This is due to the fact that...' by 'This was shown by clamping the pHi at 6.7 using artificial sea water containing ammonium acetate, a condition which did not block Cdc2/Cdk1 activation.'.

Comment 10: line 113 remove 'to'

Answer to Comment 10: As indicated, I removed 'to'.

Comment 11: line 123 ; this is why oocytes at this stage are called 'immature'.

Answer to Comment 11: As indicated, I used “this is why oocytes at this stage are called 'immature'.”

Reviewer 2 Report

Dear Authors

The reported manuscript is a review dealing with oocyte maturation in starfish.

The manuscript is well written and correctly organized. Please pay attention to the reference style reported in the journal guidelines. The topic is interesting and attractive for a broad range of audience, however for the sake of clarity, my syggestion is to improve the discussion on fish hystology.

Recent literature describs a strong relation between oocyte development and GPE cellular protection, then it could be useful to describe a little bit about this point.

For your convenience, take a look to the following paper:

Cacciatore et al. 2012 Mini reviews in Med chem 12, 13-23.

Overall my recomandation is minor revision.

Best regards

Author Response

I am very delighted to hear recommendation for publication from reviewer 2.

I replicated the comments of the Reviewer 2, and I wrote answers for the comments.

Comment 1: The topic is interesting and attractive for a broad range of audience, however for the sake of clarity, my syggestion is to improve the discussion on fish histology.

Answer to Comment 1: Thank you very much for your positive comment. But I am afraid that I did not discuss fish histology in my manuscript and that I cannot modify it. I hope that the reviewer accepts this.

Comment 2: Recent literature describs a strong relation between oocyte development and GPE cellular protection, then it could be useful to describe a little bit about this point. For your convenience, take a look to the following paper: Cacciatore et al. 2012 Mini reviews in Med chem 12, 13-23.

Answer to Comment 2: Thank you for your suggestion. I read “Cacciatore et al. 2012 Mini reviews in Med chem 12, 13-23”, but I cannot fund a paper studying relation between oocyte development and GPE cellular protection. I also checked PubMed using key words such as “GPE” and “oocyte”, but I could not find any paper. Then, I cannot modify my manuscript. I hope that the reviewer could accept this.

Reviewer 3 Report

In this manuscript, the author summarized progresses of studies on oocyte maturation in starfish mainly done by their laboratory. Also discussed about difference between mammals and starfish. Knowledge from the studies was concisely summarized. Contents are up-to date and well written.  Thus this review will give appropriate information.  I could not find out any problem and typographical error. Thus this reviewer suggests that the manuscript will be accepted as in present form.

Author Response

I am very delighted to hear recommendation for publication from reviewer 3. Thank you very much.